# Excess Winter Mortality (EWM) as a Dynamic Forensic Tool: Where, When, Which Conditions, Gender, Ethnicity and Age

**DOI:** 10.3390/ijerph18042161

**Published:** 2021-02-23

**Authors:** Rodney P Jones

**Affiliations:** Healthcare Analysis & Forecasting, Wantage, Oxfordshire OX12 0NE, UK; hcaf_rod@yahoo.co.uk; Tel.: +44-(0)7890-640399

**Keywords:** excess winter mortality, influenza, latitude, gender, age, respiratory conditions, spatiotemporal effects

## Abstract

To investigate the dynamic issues behind intra- and international variation in EWM (Excess Winter Mortality) using a rolling monthly EWM calculation. This is used to reveal seasonal changes in the EWM calculation and is especially relevant nearer to the equator where EWM does not reach a peak at the same time each year. In addition to latitude country specific factors determine EWM. Females generally show higher EWM. Differences between the genders are highly significant and seem to vary according to the mix of variables active each winter. The EWM for respiratory conditions in England and Wales ranges from 44% to 83%, which is about double the all-cause mortality equivalent. A similar magnitude of respiratory EWM is observed in other temperate countries. Even higher EWM can be seen for specific respiratory conditions. Age has a profound effect on EWM with a peak at puberty and then increases EWM at older ages. The gap between male and female EWM seems to act as a diagnostic tool reflecting the infectious/metrological mix in each winter. Difference due to ethnicity are also observed. An EWM equivalent calculation for sickness absence demonstrates how other health-related variables can be linked to EWM. Midway between the equator and the poles show the highest EWM since such areas tend to neglect the importance of keeping dwellings warm in the winter. Pandemic influenza does not elevate EWM, although seasonal influenza plays a part each winter. Pandemic influenza and changes in influenza strain/variant mix do, however, create structural breaks in the time series and this implies that comparing EWM between studies conducted over different times can be problematic. Cancer is an excellent example of the usefulness of rolling method since cancer EWM drifts each year, in some years increasing winter EWM and in other years diminishing it. In addition, analysis of sub-national EWM in the UK reveals high spatiotemporal granularity indicating roles for infectious outbreaks. The rolling method gives greater insight into the dynamic nature of EWM, which otherwise lies concealed in the current static method.

## 1. Introduction

Excess winter mortality (EWM) compares the average deaths in the four winter months against the average in the eight non-winter months [1]. The study by Johnson and Griffiths using data for England and Wales established the EMW calculation, using March as a reference point [1]. It was introduced as a standard method because up to that point various alternative methods had been used to study EWM, which hindered comparison of international and time series analysis. It is a widely studied topic with implications to health policy, insulation standards in buildings and fuel costs [2,3,4,5]. However, a recent study has introduced the use of a rolling EWM calculation because it was noted that the EWM calculation does not always reach a maximum in March [6], and that the rolling EWM calculation can also be applied to sickness absence and other health data, which typically peak before deaths, i.e., illness preceding death [7]. A rolling EWM calculation compares the average deaths in September to December versus the average deaths in January to August as a percentage difference. Move forward one month and repeat the rolling calculation. This gives a dynamic rather than a static view of EWM.

EWM has been used in multiple studies to demonstrate the importance of keeping dwellings warm in the winter [2,3,4,5]. A comprehensive US study conducted at county level used combined spatial variation in the energy source used for home heating and temporal variation in the national prices of natural gas versus electricity to investigate this relationship [8]. A lower heating price reduced winter mortality, driven mostly by cardiovascular and respiratory causes. Circulatory to respiratory mortality rate was 1.4:1. Average heating degree-days (HDD) and average age-adjusted mortality rates were highly correlated. Female EWM was markedly higher January to March [8]. However, the price of natural gas peaked in 2008 and declined thereafter as fracking increased domestic production [9]. Hence, EWM trends in the US and elsewhere will be subject to hidden trends due to the price of natural gas [9]. Many countries have recognized the need for better home insulation and in New Zealand, homes were retrofitted with insulation with resulting significantly fewer (11%) hospital admissions in the intervention population. Effects were more pronounced for respiratory disease (−15%), asthma in all age groups (−20%), and ischaemic heart disease in those older than 65 years (−25%) [10]. In theory, improved housing insulation may therefore be acting to slowly reduce EWM over time.

The study mentioned above in the US identified higher female EWM [8] and a UK study conducted in 1995 to 1998 for persons aged 75+ from across the UK established that female winter mortality was 11% higher than males and that persons with self-reported respiratory illness had a 20% higher winter mortality. Deprivation had no observable effect, however, difficulty in keeping the house warm led to a 14% increase [11].

To establish if there were multiple causes of EWM, a large study of 14 European countries from 1988 to 1997 demonstrated that more affluent countries had lower EWM, within a country poverty and income inequality elevated EWM, levels of cavity wall insulation, double glazing and floor insulation all decrease EWM, obesity and smoking did not affect EWM. Higher per capita healthcare and public health expenditure had a marked effect on reducing EWM [4]. Clearly there are multiple and interrelated causes behind short- and long-term changes in EWM.

However, there has been little study into year-to-year variation, and in countries outside Europe. This study will employ a rolling EWM calculation to assess EWM wider afield than just temperate areas, and especially investigate gender and detailed age differences in EWM. In addition, it will seek to determine if winter infectious outbreaks play a greater role and how these may interact with temperature. Both monthly and weekly data will be employed to investigate timing differences in EWM between different locations, with weekly data better able to illustrate such timing differences. The key point of this study is to demonstrate that adapting a standard method can yield far greater insight into the dynamic interrelationships behind EWM.

## 2. Materials and Methods

Monthly deaths for various countries were primarily from the United Nations [12], and Eurostat [13], plus additional data for the USA [14], Canada [15], Germany [16], Singapore [17], and Michigan [18]. These were supplemented (when necessary) from the websites of individual countries. When monthly data was (very infrequently) missing in a time series, annual data was obtained and apportioned across each month using the monthly average each side of the missing data. These averages were then adjusted to match the annual total. This mainly affected some of the smaller world countries. This does not affect the calculation of the median EWM and prevents the gap in EWM on either side of the missing data.

Data for weekly deaths in England and Wales was from the Office for National Statistics [19], as was data relating to the effect of social deprivation decile [20], and monthly deaths by age and sex [21,22]. Data on NHS sickness absence was from NHS Digital [23].

The latitude of countries/states was from www.geodatos.net (accessed on 5 January 2021).

In the original static EWM calculation of Johnson and Griffiths [1], EWM was calculated as the average of December to March versus the average of August to November before, and April to July after. In the revised rolling method EWM simply runs January to August versus September to December, move forward by one month and repeat. On some occasions only weekly data on deaths was available and the EWM calculation then becomes 17 weeks versus 35 weeks, which is the closest approximation to monthly EWM calculation.

All data was stored and manipulated in Microsoft Excel (Microsoft Corporation, Redmond, WA, USA). For countries with more than 100 but fewer than 1000 deaths per annum the skew in Poisson small numbers implies that selecting the maximum EWM each year can generate statistical artefacts which are too high. Any EWM greater than 70% was removed from the calculation of the median, and sequential deletion of high values was sometime required to give a median value comparable to other countries at the same latitude.

## 3. Results

### 3.1. A Rolling EWM Calculation versus the Original (Static) Method

However, it is important to establish if the two methods give substantially different values. Using data for England and Wales, the simple answer is that on most occasions the two methods are close to each other, with a maximum percentage point difference of ±3%. The rolling EWM method is therefore reliable and will give consistent answers when EWM reaches a maximum in different periods, as is the case for countries closer to the equator.

For example, data from England and Wales from the winter of 1952/53 through to 2018/19 shows that while EWM reaches a maximum in March in most years, this is not exclusively so, with a maximum in January in 1989/90, 1993/94 and 2003/04, and in February in 1995/96. While males and females usually peak together in the winter of 1949/50, EWM reaches a maximum in March for males but April for females, in 1951/52 in March for males and April for females, and in 2010/11 for males in January and females in March.

Clearly March does not apply to the Southern hemisphere. Data for Australia shows that while EWM mostly peaks in September it can also do so in August and October. The date for maximum EWM in both England and Australia becomes more variable at the regional or state levels.

Nearer to the equator in Singapore (latitude 1.3° N) EWM can peak anywhere in the year, but mostly in June and August (see Figure 1). The March peak corresponds to the first of two influenza seasons, which has a weaker Northern hemisphere emphasis while the later peak has a stronger Southern hemisphere emphasis. Nearer to the equator where temperatures are similar throughout the year EWM more correctly becomes excess influenza season mortality.

A rolling EWM calculation is therefore the preferred option when looking to investigate EWM wider afield than just Europe. As opposed to the static method it gives the genuine maximum EWM figure in each year. Investigation of the international data showed that selecting the maximum value of EWM in January to December of each year encapsulates the true maximum in all countries.

### 3.2. Role of Latitude

Latitude is known to influence EWM [4] and this is investigated in greater detail for 130 countries. The EWM is the median value across the years available for each country. Most European countries have close to 59 years data, but the median value is 29 years. In some countries EWM shows higher values in the interval 1960 to 1970 and these have been removed from the time series. Countries with fewer than 100 deaths per annum were also excluded.

The resulting relationship is shown in Figure 2 where the median EWM reaches a maximum at 35 degrees latitude, which is midway between the equator and the poles. EWM reaches a minimum near the equator of around 5% and another minimum at 65 degrees latitude of around 9%. The scatter around the trend line indicates that country specific factors are also involved. The tropics of Cancer and Capricorn are 23 degrees from the equator. The highest median EWM values are for Malta 32%, Portugal 28%, Gibraltar 26%, Israel and Cyprus 24%, Macao (China) 21%, and Ireland and Spain 20%. Most of these have a Mediterranean climate.

### 3.3. Role of Respiratory and Other Conditions

Higher respiratory deaths in winter are a common observation [24]. However, EWM calculations are infrequent. Such a rolling calculation is given in Figure 3 using weekly data for England and Wales.

As can be seen, EWM for respiratory conditions varies between 44% in the winter of 2011/12 to a massive 83% in the winters of 2014/15 and 2017/18. The larger the winter peak, the greater the corresponding dip in respiratory deaths the following summer. This is an example of the culling effect where death is slightly moved forward in time due to adverse conditions. Both respiratory and all-cause deaths peak at the same time, mainly because it is respiratory deaths which are driving the marginal changes in all-cause deaths.

In the worst winters’ respiratory deaths (in the worst week) can account for 20% of all-cause deaths. In less adverse winters this can drop to 15%. Note the different shapes of the rolling EWM calculation which reflect differences in timing and magnitude across the smaller parts of England and Wales. The largest peaks seemingly correspond to highest influenza activity as an expression of a wider mix of winter pathogens.

Data from the state of Michigan in the USA contains monthly deaths for various conditions [18] which allows calculation of a rolling EWM wider than just all respiratory conditions. These results are shown in Table 1 where the maximum EWM (2000–2020) is shown, including when this maximum occurred and the proportion of total deaths was due to the condition.

As can be seen in Table 1 pneumonia and influenza achieve a higher EWM that respiratory conditions in general as seen in Figure 3 for England and Wales. The next highest are septicemia and COPD at around 40%, followed by heart disease and stroke at 19%, etc. The all-other group is slightly higher than for all deaths.

Cancer represents a special case since the rolling EWM calculation shows that it is subject to systematic undulations which do not exactly correspond to a winter cycle. For example, maxima in October-01, October-02, January-04, January-05, August-05, October-06, January-08, January-09, January-10, January-11, October-11, etc. Note the differences in timing for maximum EWM in each category. Due to the relative proportion of all deaths a mix of heart disease, all other and cancer will drive the EWM for all deaths, with the systematic undulations for cancer acting to increase or decrease EWM in specific winters. See Section 4.9 for discussion.

### 3.4. Role of Age and Gender

Roles for age and gender are becoming increasingly recognized as important factors in EWM [8,11]. Table 2 demonstrates the importance of age in terms of the maximum EWM and the volatility in EWM (as the standard deviation). Maximum EWM for each age reaches a minimum at age band 15–44 years, as does the standard deviation.

A lower standard deviation implies that EWM is less sensitive to winter. The higher values at a young age looks to be a new discovery since it has been assumed that EWM was restricted to the elderly. Female EWM is higher than male in all age bands.

While such an overview is useful it would be helpful to see if finer age graduations can be discerned along with additional gender-specific interactions. As can be seen in Figure 4, median female EWM is higher than the male through to age 35–39, is then only slightly above male EWM up to age band 55–59, then goes above that for males until male EWM goes higher for age 90+.

However, dynamic analysis shows a far more complex set of age-gender relationships, shown in Figure 5. The magnitude of the gender difference is constantly changing with time for each age band; hence, age 70–74 reaches a maximum gap of 7% in March 2007, while age band 85–89 reaches a 7% difference in March 1997. The apparent lower EWM for males aged 90+ in Figure 4 comes from a run of low male EWM from 2004 to 2010, which is not seen in other years. Further age-specific peaks occur each winter.

It is for this reason that it has been suggested that the gender difference reflects the mix of pathogens and metrological conditions operating each winter, and their interactions [25,26]. EWM is a vastly complex system of interactions.

### 3.5. Effect of Location

Regional differences in cold-related risk are well documented [27]. Table 3 shows the variation in maximum EWM, and when this occurs between regions in England and Wales. The standard deviation is that associated with the rolling calculation. There is no relationship with latitude and other factors can be assumed to be of greater influence. The differences in the standard deviation are statistically significant since from Poisson statistics the largest region should have the smallest standard deviation, and the smallest region the largest. Spatial spread is implied by the timing differences for the maximum.

### 3.6. Role of Area Deprivation Score

Previous studies have not demonstrated a significant role for deprivation in EWM [28]. This issue is explored using the small area Index of Multiple Deprivation (IMD) used in the UK [29]. IMD is then divided into deciles with decile 1 being most deprived and decile 10 being least deprived. As before, no relationship with deprivation can be discerned (Table 4). However, of greater interest is the variation in the date at which the maximum EWM value was observed. While March 2015 and 2018 are common in both England and Wales, Wales has a higher proportion in 2018 and an April 2013 date for the least deprived areas.

Recall that Table 4 gives the maximum EWM which will mostly occur during influenza outbreaks. While varying deprivation is spread across all local authority areas in England and Wales, high deprivation in Wales usually occurs in the Welsh Valleys which are former mining areas and in parts of Northern England. Least deprived areas of Wales are usually in or near larger cities [30]. Spatiotemporal spread of infectious agents is suggested to account for these timing differences—see Section 3.12.

### 3.7. Roles for Ethnicity

Differences due to ethnicity seem not to have been investigated as a cause of dynamic behavior in EWM. Figure 6 explores this issue using data from Singapore where monthly deaths are reported by ethnicity. Chinese are the predominant ethnic group followed by Malays at around 4:1. In Figure 6 the difference in EWM is calculated between Malays and Chinese. The absence of any ethnic differences should generate random scatter; however, the trend shows systematic patterns.

Hence in August 1961 Malay (M + F) EWM is 10% (percentage point difference) higher than Chinese while in July 1969 it is 15% lower. During the Hong Kong influenza pandemic which commenced in 1968 the Malays experience 3% higher EWM than the Chinese. Temperature can be excluded as a causative factor in Singapore. These differences can be explored between males and females and hence Figure 6 also shows the difference in EWM between Malay and Chinese females. The spread in the pattern for females will be slightly exaggerated due to smaller number Poisson variation [31], however, differences in timing and magnitude are evident. The female difference is especially marked in July 1962, November 1963, etc., and Malay females shower higher EWM during the Hong Kong flu pandemic which commenced in 1968 in the far East. EWM is seemingly far more nuanced than has been thought possible.

### 3.8. Factors Affecting the Median Value of EWM

Most of the studies on EWM calculate an average or median value over a range of years, which is then used as a basis for further analysis. Does the date range make a material difference? Figure 7 has already given one example of where the date range did matter for females/males aged 90+. These issues are explored further in Figure 7 where a rolling 3-year average is applied to EWM data for Singapore. Recall that in Singapore temperature variation is minimal because it is so close to the equator. As can be seen there are indeed clusters when EWM is significantly higher or lower. Hence the period 1965 to 1973 is consistently high while 1998 to 2003 is consistently low. Having dismissed temperature as a factor it can only be concluded that the frequency and magnitude of infectious outbreaks are the main cause of this behavior in Singapore. Figure 8 explores this issue in greater detail for more temperate areas of Europe and gives a time series from 1960 to 2019 for three European countries. Germany as an example of an affluent country, Sweden of a Scandinavian country with buildings designed to cope with cold winters, and the United Kingdom with gradients in wealth and a legacy of poor buildings unsuited to cold winters.

There are several important points to be derived from this figure. Firstly, as expected, Sweden has the lowest average EWM due to its higher building standards designed to cope with cold winters [4]. Germany is lower than the UK due to its higher relative affluence [4]. However, there are peculiar features associated with each country. For example, in March 2012, Sweden recorded the highest EWM of the three countries. The UK appeared to have high EWM from 1960 to 1970, then suddenly dropped to a lower level which continued until March 2000, when another drop is observed.

As opposed to the UK, Sweden had a run of low EWM from 1960 through to 1975. Like the UK, Germany was high from 1960 through to March 1970 when it also showed a sudden decline. Germany had a run of low EWM from 1977 to 1984. Such country-specific patterns make comparisons between different studies somewhat more difficult. Seemingly everyone concentrated on the average and ignored the variation.

This observation leads to the interesting possibility of ranking countries according to the stability of their rolling EWM. Countries with the lowest standard deviation (highest stability) are at the top of Table 5 while those which are most volatile are at the bottom. Unsurprisingly, all the Scandinavian countries are at the top of the table while countries with known housing standard issues are toward the bottom. If influenza outbreaks are a part of year-to-year volatility, then it must be concluded that colder indoor temperatures in the winter simply exacerbate influenza mortality—but perhaps do not influence its incidence.

### 3.9. Pandemic versus Seasonal Influenza

In the field of influenza epidemiology, it has been noted that pandemic influenza with a substantially different strain does not seemingly greatly change influenza attributed mortality [32]. A peak in deaths due to influenza will be reflected in the timing of the maximum EWM calculation. Figure 9 shows the difference in EWM for the winter of 1967/68 versus that in 1968/69 during the Hong Kong Flu (H3N2) pandemic.

Data for Singapore peaks in September 1968 and seem to be earlier and of higher magnitude than other locations due to its proximity to the epicenter of the outbreak. However, on most occasions the earlier seasonal influenza outbreak in 1967/68 had a higher magnitude that for Hong Kong flu. Both trend lines are second order polynomials. The median value of EWM since 1960 in Singapore is around 7%. The Hong Kong flu arrived in Europe during the winter of 1968/9 and, compared with countries in Europe, the EWM during the 1968/69 winter was almost universally lower than the previous winter.

### 3.10. Deaths and Sickness Absence

It has been suggested that illness will precede death and hence a rolling calculation of excess winter sickness absence (EWSA) should peak before the corresponding EWM [7]. This is illustrated in Figure 10 where both EWM and EWSA are calculated for the Midlands of England. On this occasion sickness absence is for NHS staff who will have a lower average age than for EWM; however, it illustrates the principle. As can be seen EWSA typically peaks 1 to 2 months prior to EWM and has a slightly different shaped profile.

The data at the far right is during the COVID-19 epidemic, during which deaths have been far higher than NHS staff sickness absence. The downward movement of EWM and EWSA is interrupted by the arrival of COVID-19 in late March 2020. EWSA and EWM are generally higher together, although not universally so. The winter of 2013/14 is far higher for sickness absence relative to deaths than any other winter. However, the point has been demonstrated that an EWM style calculation can be applied to other health related variables such as medical admissions, etc.

### 3.11. Place of Death

It is possible that place of death may influence EWM since locations such as nursing homes will have a high proportion of frail individuals. Table 6 explores this possibility using EWM for the 2017/18 winter in England and Wales. Place of death relates to the institution where death occurs, hence it is not necessarily the same as where you live. i.e., only 24% of all deaths occur at home.

As can be seen, residents of care homes (which includes nursing homes) have the highest EWM, followed by communal establishments, which will mainly be university halls of residence and boarding schools (recall the peak in EWM for young adults in Figure 4), followed by hospital, elsewhere (usually persons away from home on holiday, recreation, etc.), then home and finally hospice (mainly terminally ill cancer patients). Note the timing differences with hospice residents (those who are weakest) showing an earlier peak, and elsewhere the latest to peak (those active enough to be away from home). The overall winter EWM will be driven by those with the largest proportion of deaths, namely, hospital, care home and home. See Section 4.9 regarding the wider context and discussion. EWM and cancer were highlighted in Section 3.3.

### 3.12. Sub-National Behavior

Much of the above analysis has hinted at the spread of infectious agents during the winter months, which is known to show high spatiotemporal granularity [25,26]. If this is correct, then the sub-national EWM should show the greatest differences during the winter months. This possibility is explored in Figure 11 where the interquartile range (IQR) is given for the rolling EWM calculation in 510 UK local authorities, counties, clinical commissioning groups, and regions. As can be seen the IQR of the rolling EWM calculation generally reaches a minimum around September or October and climbs to a maximum around January to April.

Recall that a maximum IQR implies high spatiotemporal granularity with some areas affected far worse than others. Also note the differences in the shape and amplitude of the IQR over time. The large peak in IQR for the winter of 2014/15 is reflected in already reported extreeme variability at small-area level, i.e., sub-local authority level [26].

The large truncated peaks during 2020 are due to the spread of COVID-19 throughout the UK during the first and second waves. The IQR reached a maximum of 17.9% in May 2020. The spread of COVID-19 is known to show massive spatiotemporal granularity [33]. Had it not been for the arrival of COVID-19 the winter of 2019/20 would have had the lowest IQR for any winter since 2001/02. Hence the national value of EWM conceals local area infectious spread as one of the primary drivers of changes in the magnitude of EWM each year.

Using an international data set of 84 countries from this study with data available in the 1980s gives an IQR of 24% during the winter of 1984/85, indicating extreme international differences during what was the largest influenza outbreak seen in the USA since the winter of 1975/76 [34]. During this winter, EWM reached 30% in Bulgaria but only 10% in Canada and Iceland, which exemplifies the high spatiotemporal variation of influenza outbreaks.

## 4. Discussion

### 4.1. The Rolling EWM Method

While heating degree days (HDD) is recognized as a better method to estimate cold-related deaths [35] this study is about EWM in general, irrespective of cause. The benefits of the rolling method for EWM lie in the fact that monthly deaths are readily available for most countries and that the rolling method involves only simple spreadsheet manipulation. This study has demonstrated its use in tropical as well as temperate regions and in giving greater insight into the dynamic aspects of EWM.

### 4.2. Role of Latitude

While EWM has been extensively documented in the more temperate countries, there is little data near the equator and this study fills this gap. As expected EWM declines to a minimum near the equator (Figure 2), however, due to the absence of seasonal temperature gradients the measure of EWM is more correctly a measure of influenza outbreaks. As mentioned for Singapore, there can be two periods of influenza as the equatorial zone is exposed to both Northern and Southern hemisphere outbreaks. A study conducted in the states of Brazil which span the equator and past the Tropic of Capricorn suggested that influenza amplitude from a Fourier decomposition may reach a minimum at 15°, however the scatter between the equator and 15° was remarkably high [36]. The scatter in Figure 2 at such latitudes is also high and such a minimum is possible. Other climatic factors may account for this scatter. In Brazil, the timing of the major seasonal peak was far more variable between the equator and 15° which has similarities to Singapore in Figure 1. This illustrates the potential danger of attempting to use data from large countries such as the USA, China, Russia, etc.

The high values of EWM midway between the equator and the poles has already been noted and is largely due to lack of thermal insulation and resulting difficulty in keeping houses warm in winter [3,4,35,36,37,38]. High EWM in the UK and Ireland are due to the same cause [3,35,37,38]. EWM then declines toward the poles due to thermal design of buildings [3,4,35,36,37,38]. The equation in Figure 2 can be used to adjust EWM from different latitudes should this be required.

### 4.3. Respiratory and Other Conditions

It has been repeatedly observed that respiratory conditions show the greatest sensitivity to winter and EWM. Hence, in New Zealand an EWM of 83% for ICD-10 Chapter X (Respiratory), and 68% for Chapter I (Infection), and 47% for symptoms, dropping to 29% for diseases of the skin [39]. The range of 44% to 83% in England and Wales (52.3 North), Figure 3 is therefore broadly consistent with New Zealand (41.2 South) [39], despite differences in timing between the two studies. See Section 4.8 for more details. The wide range observed in Figure 3 is almost certainly due to higher levels of influenza and associated winter pathogens in some years which mainly affect the respiratory system. This then makes a large contribution to the overall EWM.

Data from Michigan (USA) gave further insight of influenza/pneumonia showing the highest EWM followed by septicemia and COPD (a respiratory disease). The other conditions group was close to that for all deaths. A curious pattern from cancer then acted to modify the all-deaths trend. See Section 4.9 for discussion.

### 4.4. Age and Gender

Roles for age and gender have been recognized for some time. For example, in New Zealand the young and elderly were recognized to experience higher EWM, while females had 9% higher EWM than males [39]. This study has gone beyond broad age bands, as in Table 1, and explored the effect of age using 5-year age bands by gender (Figure 4 and Figure 5). One of the key observations is for extremely high female EWM around the age of puberty (age 10–14) which requires more detailed examination by cause of death. However, for adults EWM increases above age 45 and keeps rising with age.

The difference between females and males by age band (Figure 5) reveals complex time trends which are discussed further in Section 4.8 dealing with EWM and influenza. Evidence for curious trends/patterns due to gender have been reported for the costs of long-term conditions in the USA [40], and more recently for admissions relating to a wide range of diagnoses [25]. These have relevance to deaths by virtue of the nearness to death effect where hospital admissions and costs escalate in the last year of life. See Section 4.7 and Section 4.8. However, such curious trends are usually dismissed because they do not neatly fit with currently accepted notions of how health care trends are supposed to behave.

Clearly the distribution of age at death between different locations, and the changes in age at death overtime have the potential to modify the overall EWM figure. Calculation by the author shows that the ageing population does not make a huge contribution to EWM trends.

### 4.5. Location and Area Deprivation

Regional differences in cold-related risk are known to occur [27]. In New Zealand regional differences in EWM were minimal [39], although are higher in Portugal [37] mainly due to the ability to afford winter heating. EWM and cost of heating were also linked in the US [2,8]. There is almost universal agreement that housing standards make the greatest contribution to *average* EWM [3,4,37,38], followed by the ability to pay for heating [2,3,8,10,37]. The same also holds for winter hospital admission rates [10].

Slight regional differences were observed in Table 2 between the maximum EWM, the volatility in the EWM (as a standard deviation), and the timing of maximum EWM. Spread of infectious agents such as influenza(s) looks likely. Similar timing differences were seen in Table 3 which grouped areas in England and Wales by deprivation decile. Similar timing differences have been observed between US and German states and UK local authorities [25,26,41].

In the UK, a winter fuel payment was introduced in 1997 to all persons of pension age [42]. A structural break in the EWM time series in the UK after the winter of 1999/00 was interpreted in one study to be due to the winter fuel payment [43]. However, the timing of the structural break is slightly out with the timing of the fuel payments and the structural break at 1999/00 also holds more widely around the world [6] and was due to a drop-in influenza activity. Hence, the winter fuel payment may not have made a substantial impact on EWM.

The lack of a relationship between EWM and area deprivation score in Table 3 is consistent with other observations that deprivation per se does not play a role in EWM [11,39]. This is possibly because in the UK persons receiving social security payments tend to live in social housing which has higher building standards that many owner-occupied or private rented properties [44].

### 4.6. Ethnicity

Ethnicity is not widely studied in EWM. A study in New Zealand concluded that there was no difference [39]. The study based on data for Singapore therefore gives useful insight into the role of ethnicity in the influenza-related part of EWM. Figure 6 focusses on the gender differences concealed in the rolling EWM trends. The black EWM line presents the raw EWM trend, the green line shows the difference between Malays and Chinese, while the red line shows the difference between Malays and Chines for females alone. The date range 1960 to 1974 was chosen to allow the trends to become clearer to view.

These subtle differences in magnitude and timing would otherwise go unnoticed. Influenza is not the only pathogen affecting human health and the trends will be the composite picture of the interaction between all pathogens, which will have a more tropical disease emphasis in Singapore.

Over many years life expectancy of Chinese has always been highest, while that of Malays has remained lowest [45]. These are probably reflected in income and socioeconomic situation. The Malays could therefore be assumed to experience a higher burden of infectious diseases, although the types may vary between the two groups.

More research is needed to determine if this is due to ethnicity or socioeconomic position. However, the male female differences do appear to be important.

### 4.7. Age or Nearness to Death

A potential role for nearness to death rather than age was identified in a French study where nursing home residents showed the highest EWM [46]. It has been observed across multiple countries that in the last six months of life, irrespective of age, people experience rapid deterioration in health resulting in escalating hospital admission (see references in [25,26]). This led to the suggestion that influenza vaccination may become less effective in this group [26] since they are already far more susceptible to any agent precipitating final demise. Extrapolating the age-dependent vaccination effectiveness rates against influenza A(H3N2) during the 2012/13 season [47] shows that vaccine effectiveness could drop to zero somewhere around age 90 and could presumably go negative above this age. This is not an argument against influenza vaccination, only that its benefits lie mainly among those not in the last year of life. Alas, this can only be retrospectively determined and may include some preventable deaths from influenza. Disentangling the two would appear to be a complex problem.

### 4.8. EWM and Influenza

There are only three primary reasons for EWM to vary each year in different countries, namely, temperature per se, temperature fluctuations (cold snaps) and winter infectious outbreaks of which influenza is a key pathogen. Periods of cold dry air are known to elevate both pneumonia and influenza mortality [48]. Figure 9 has demonstrated roles for influenza outbreaks in the variation of EWM. After the Hong Kong flu (H3N2) had struck Singapore, two variants had emerged which then infected Europe and the North Americas [49], and this may explain why Singapore showed such a large earlier response. The effect of influenza epidemics/pandemics seems to entirely depend on the degree of antigenic similarity to similar strains which the elderly has been exposed during childhood and through life [50,51]. Hence, changes in influenza strains, vaccine effectiveness and proportion of elderly vaccinated should all interact with temperature to change the magnitude of EWM.

The joint role of influenza and temperature can be illustrated in England and Wales during the winter of 1950/51 when EWM rose to 68.2% in males and 70.7% in females in March 1951 (authors calculation). The winter of 1950/51 was the coldest in 100 years and was experienced across most of the Northern Hemisphere [52]. This winter was so ferocious that military operations during the Korean war were substantially affected [53]. This winter also involved a substantial outbreak of influenza A(H1N1) [54].

Recall that in the 1950s most homes in the UK did not have central heating and insulation was almost absent. Heating was from coal fires which would have created additional lung irritation. However, the principle has been established that extreme cold plus influenza leads to substantial elevation in EWM. A study in France showed that age 75+, resident in a nursing home, monthly minimum temperature, and influenza activity (as influenza-like-illness) gave good prediction of EWM [46].

In the US, the peak months of mortality for ischemic heart disease, cerebrovascular disease, and diabetes mellitus coincided with peaks in pneumonia and influenza. The magnitude of the seasonal component was highly correlated with measures of excess mortality and was significantly larger in seasons dominated by influenza A(H2N2) and A(H3N2) viruses than in seasons dominated by A(H1N1) or B viruses [55].

There was an age shift in mortality during and after the 1968/69 pandemic in each disease class, with features specific to influenza A(H3N2) [55]. This study demonstrated that there were increases in the proportion of excess pneumonia and influenza mortality that occurred in the younger age group (<75 years) during the 1968/69 pandemic year. The magnitude of the age shift decreased steadily until 1975/76, after which time the age shift appeared to be independent of the circulating influenza virus type but continued to move toward older age groups [55]. This is highly relevant to shifts in EWM discussed in the Section 4.9.

In Portugal, influenza has also been associated with higher EWM [56]. Approximately 90% all-cause deaths in the winter were for those aged 65+. Excess mortality was 3–6 fold higher during seasons dominated by the A(H3N2) subtype than seasons dominated by A(H1N1)/B. High excess mortality impact was also seen in children under the age of four years. Seasonal excess mortality rates from cerebrovascular diseases, ischemic heart diseases, diseases of the respiratory system, and chronic respiratory diseases, were highly correlated with each other and with seasonal rates of influenza-like-illness (ILI) among those aged 65+ [56].

Singapore therefore provides a useful benchmark into the role of influenza in the absence of winter cold. Apart from a 19% value for EWM during the Hong Kong flu, maximum EWM has never risen above 12% to 13% from 1960 through to 2020 (13% in July-62 and August-06, and 12% in June-97 and August-17), and the median value is only 7%.

Subtracting EWM in Singapore from EWM in the UK yields a 28% (percentage point) gap during the Hong Kong flu, 21% in the winter of 1999/00, 19% in 2017/18 down to 1% in 2005/06. The actual mix of strains and variants between the two will differ but cold seems to play a far greater role in the UK. This was the somewhat disputed conclusion of Donaldson and Keating in 2002 [57]. Winter humidity and rainfall were also associated with EWM across Europe [4]. Clearly winter mortality from influenza is very much influenced by climate, which is exacerbated by home insulation standards and the ability to maintain a suitable temperature. Such a notion is supported by the study of Laake and Sverre [38], who concluded that the gap between the UK and Norway was due to temperature (and its implications in less well insulated homes) and not influenza per se for those aged 65+.

There is one exception which was pointed out in a study of EWM in France, namely, residents of nursing homes [46]. Nursing homes are universally kept warm during the winter, yet the residents were observed to suffer highest EWM. This is probably a reflection of the phenomena called the nearness to death effect (NTD), where rapid decline in health generally occurs in the last year of life discussed in [25,26]. This possibility was confirmed in Table 6 where care home residents have almost double the EWM of persons living at home. Higher EWM then occurs by frailty, other than for persons dying in hospice since the terminally ill will die on a more even annual basis, or places in hospices are limited which caps the extent of EWM. For care homes, winter is therefore about transmissible winter pathogens, rather than indoor temperature. The high EWM for other communal establishments, mainly student halls of residence and boarding schools—refer to Figure 4 where adolescents have remarkably high EWM, seemingly due to traffic accidents and suicides which will peak in the winter [58].

Under such circumstances it was proposed that influenza vaccination may lose its protective effect, and the nursing home residents will succumb to whichever pathogen they first encounter, presumably bought into the nursing home by staff or visitors. From Table 6 it was observed that in England and Wales some 22% of all deaths occurred in a care home, and a significant proportion of the 47% of deaths in a hospital will also be persons from a care home.

Section 3.12 and Figure 11 illustrate the truth that all infectious outbreaks show high (sub-national and international) spatiotemporal granularity. Hence during the winter of 2014/15 in the UK, the EWM was the highest in around 25 years, however, the IQR ranged from an EWM of 17% at the lower quartile to 27% at the upper quartile. This was not an extreme winter in terms of temperature and the range in sub-national EWM therefore arose from the spread of, and interaction between, winter infectious agents, of which influenza is but one agent [25,26]. Even more curiously the gap between female and male EWM in the winter of 2014/15 (9.2% difference) was the second highest since 1949, only slightly lower than 1996/97 (9.2% difference). Influenza happens very frequently so what happened in these two years that causes such a huge difference between the genders? No pathogen ever operates in splendid isolation [26].

Finally, the wider effect of influenza vaccination on EWM requires some thought. Vaccination is widely recognized for its benefits and along with other public health measures acts to increase life expectancy, i.e., ultimate death is not prevented but simply shifted to a future date. Hence while vaccination enhances life expectancy, like all other interventions it may well fail to confer benefit in the last months of life. Hence, paradoxically it should have little effect on EWM as was observed in the French study with nursing home residents. An interesting topic for future research.

### 4.9. Intrpreting Shifts in EWM

Both Figure 7 and Figure 8 demonstrated sudden shifts in the value of EWM during a time-series. It is known that the emergence of a significant new variant of influenza (as in a pandemic) leads to a shift to higher mortality at younger ages. The study of Reichert et al. [55] demonstrated that such a shift occurred during and after the 1968/69 Hong Kong flu pandemic in each disease class, i.e., for ischemic heart disease, cerebrovascular disease, and diabetes mellitus, pneumonia, and influenza with features specific to influenza A(H3N2). There were increases in the proportion of excess pneumonia and influenza mortality that occurred in the younger age group (<75 years) during the 1968/69 pandemic year. The magnitude of the age shift decreased steadily until 1975/76, after which time the age shift appeared to be independent of the circulating influenza virus type but continued to move toward older age groups [55].

In England and Wales age-specific shifts were also demonstrated in Figure 5, although data did not go back to the 1968/69 winter. Nevertheless, both macroscopic and age-specific shifts do occur and are probably due to the emergency of new influenza variants.

However, a perusal of the time series for various European countries shows that the 1968/69 shift to lower EWM was seen in Germany, UK, Portugal, Greece, Italy, France, Ireland, Spain, Austria, and Belgium. In contrast, a shift-up appeared to occur in Sweden, while no shift or only minimal change occurred in Switzerland, Netherlands, Denmark, Finland, Iceland, Norway, and Liechtenstein. Most of these are Scandinavian countries. The relevant point is that the trends in EWM are country specific with suspected involvement of influenza strains, but that such shifts can make calculation of average and median EWM subject to distortion depending on the time scale employed in various studies on this topic.

It is worth noting that the timing and mix of influenza(s) vary considerably each year between countries and within regions of a country [59]. Hence an explanation for timing differences in maximum EWM seen in Table 2 and Table 3.

### 4.10. Interpreting the Shape of the Rolling EWM

Figure 9 shows a curious variety of shapes for both EWM and EWSA which require further elaboration. Winter and influenza are not the only factors capable of precipitating final demise. There are over 2000 known species of human pathogen which interact in extraordinarily complex ways discussed in [25]. In addition, a curious international pattern of shift-up and shift-down has been shown to apply to the trends in deaths [26]. Shift-up or -down can occur at any point in the year and at small area level behaves like an infectious outbreak with both deaths, medical admissions, sickness absence all affected. Shift-up has been shown to alter the shape of the rolling EWM [26]. In the past, these shifts were not realized because most countries analyzed mortality on an annual basis. Clearly something of importance is happening which requires further investigation.

The observation that cancer is showing curious undulations in a rolling EWM calculation is not surprising since curious long-term patterns have been observed in male/female cancer costs [40], in incidence and costs for specific cancers [60], and hospital deaths for cancer [61]. These curious patterns have been proposed to be associated with outbreaks of a new type or kind of disease. Further research is required.

### 4.11. Future Research Directions

The whole issue of age, gender, and ethnicity in EWM remains poorly studied. The high values of EWM in teenage females requires detailed examination of cause of death. The relationships are highly dynamic and seem to be mostly driven by infectious outbreaks. While influenza remains an obvious candidate the relationships between a multitude of winter pathogens are themselves competitive and dynamic [26]. Influenza vaccination in isolation does not seem to make a major impact on EWM seemingly because in the old and frail, who are most likely to die in winter, vaccination may protect against influenza but another pathogen for which there is no vaccine then precipitates final demise [26]. All these factors will benefit from wider research.

### 4.12. Strengths and Limitations

While daily and weekly deaths are preferred for more detailed studies, monthly data on deaths is readily available from all statistical agencies. Weekly data is becoming more widely available and Mortality.org has weekly deaths data for 38 countries. The rolling method means that EWM is available immediately as data is available rather than having to wait for another 4-months to elapse.

Serfling, ARIMA, Wavelet, Fourier, etc., may be preferred for precise characterization of properties, however, the rolling EWM method is easy to apply and can be used to screen for multiple causes such as age, gender, cause of death, place of death, etc., as has been done in this study. The method is also more widely applicable to other health-related properties such as sickness absence, hospital admissions, etc. For example, the rolling EWM method has been used to demonstrate why healthcare locations experience variable capacity and financial pressures at different times [25,26]. Analysis in this study used a mix of weekly and monthly data which was dictated by the availability of data already in the public domain.

## 5. Conclusions

A static view of EWM is demonstrated to give a simplistic picture of its real-world complexity. A rolling calculation acts to reveal the highly nuanced spatiotemporal, gender and ethnic behavior. Singapore was used as an example of an equatorial country where temperature could be excluded as a major variable. Results from Singapore indicate that influenza per se is only capable of 12% to 13% increases in EWM, and that anything above this must be due to a mix of temperature/metrological effects on influenza mortality. There are a series of breaks in the EWM time series which appear to coincide with pandemics or other shifts in the mix of influenza(s) and their variants. The behavior over time is therefore driven by the frequency and magnitude of infectious events of which influenza is a major contributor. Care needs to be taken in comparing average or median EWM calculated over different time periods. Respiratory conditions were shown to display extremely high EWM in England and Wales. This may reflect the generally damp winters in the UK and a legacy of poorly insulated dwellings. A rolling calculation of sickness absence using the EWM methodology showed that death lags sickness absence, i.e., illness precedes death. This is an entirely expected outcome, but demonstrates that static time frames cannot be used when seeking to link EWM to other health parameters. As has been reported previously countries mid-way between the equator and poles experience the highest EWM because they experience the highest temperature extremes and generally do not have adequate building standards suited to maintaining household temperature during the winter. This seemingly amplifies the mortality arising from winter infectious outbreaks. This may have led to over-attribution of influenza deaths in some countries. Finally, while EWM was instrumental in identifying the need for maintaining indoor temperature during the winter, EWM is a far more nuanced measure of temperature and infectious outbreaks. For these reasons it is probably not a good tool to measure the success or otherwise of national schemes to improve insulation standards or of financial aid to the elderly to pay for heating over the winter.

## Figures and Tables

**Figure 1 ijerph-18-02161-f001:**
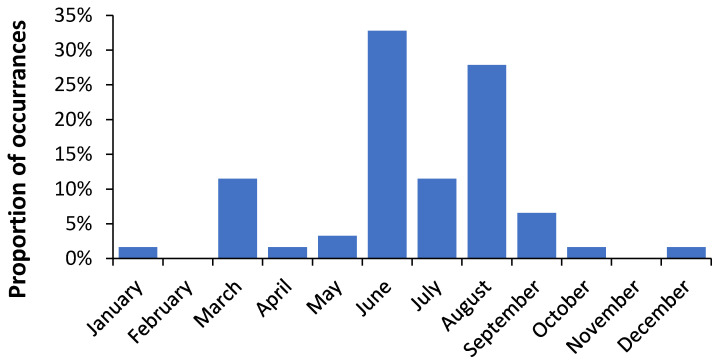
Month at which a rolling EWM (Excess Winter Mortality) calculation reaches a maximum in Singapore, 1960 to 2020.

**Figure 2 ijerph-18-02161-f002:**
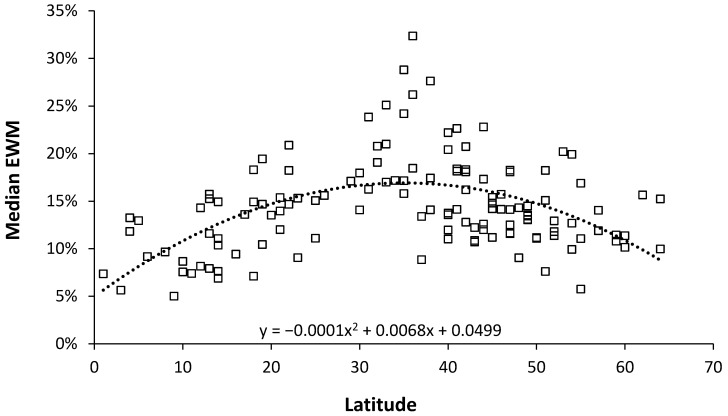
Median value of the EWM for 130 world countries versus latitude. Median for each country (□), dotted line second order polynomial line of best fit.

**Figure 3 ijerph-18-02161-f003:**
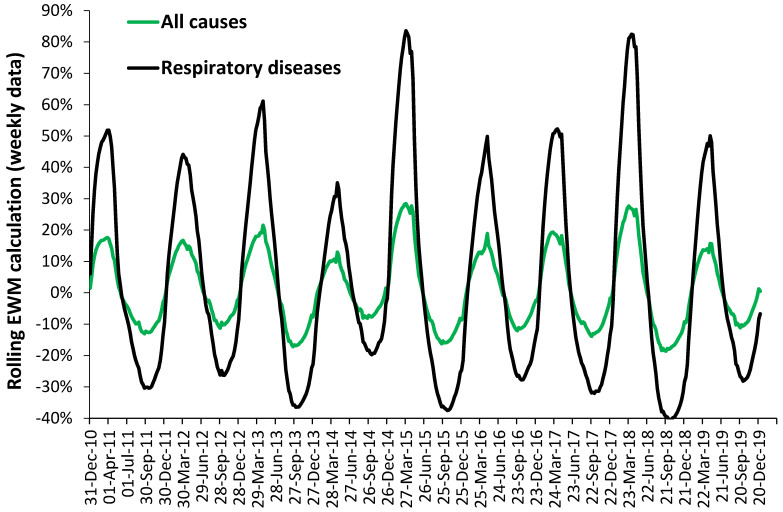
Rolling EWM calculation in England and Wales (weekly data) for respiratory diseases (ICD-10 Chapter J) and for all-cause deaths.

**Figure 4 ijerph-18-02161-f004:**
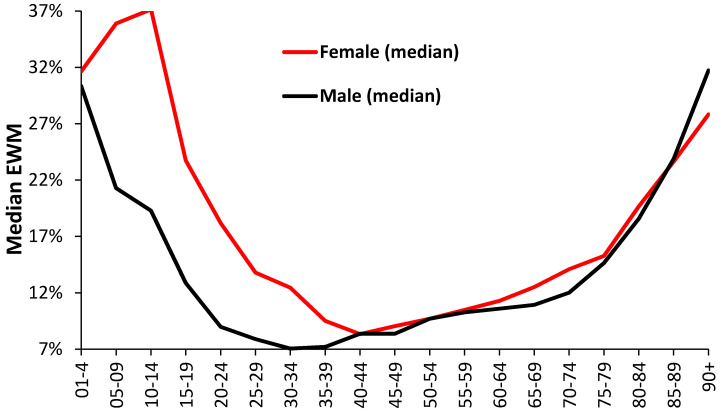
Median value of EWM (1993/94 to 2015/16) for males and females by age band.

**Figure 5 ijerph-18-02161-f005:**
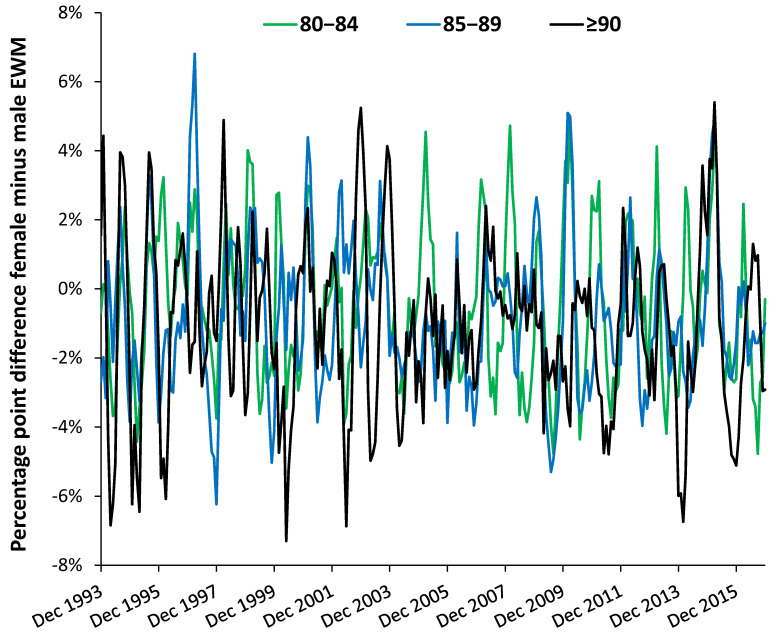
Percentage point difference between female and male EWM (1993/94 to 2015/16) for age bands over 79 years.

**Figure 6 ijerph-18-02161-f006:**
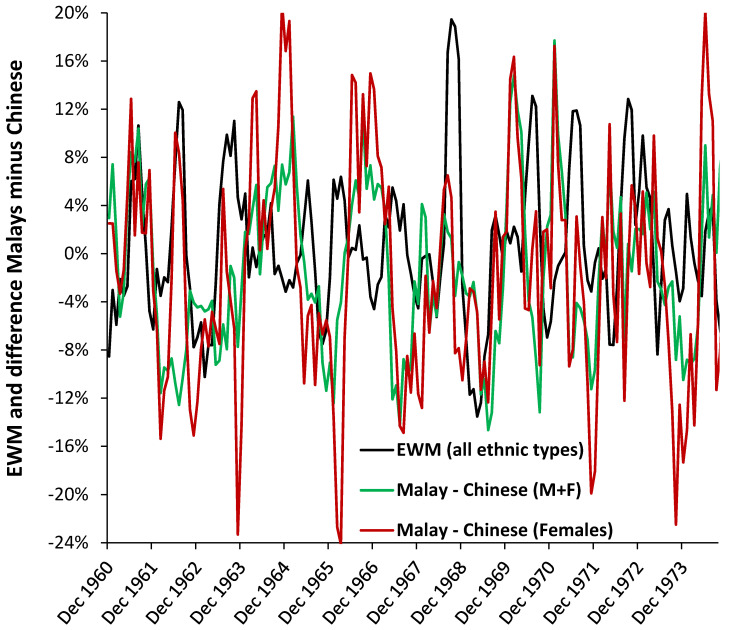
Difference in EWM between Malays and Chinese in Singapore (1960 to 1974).

**Figure 7 ijerph-18-02161-f007:**
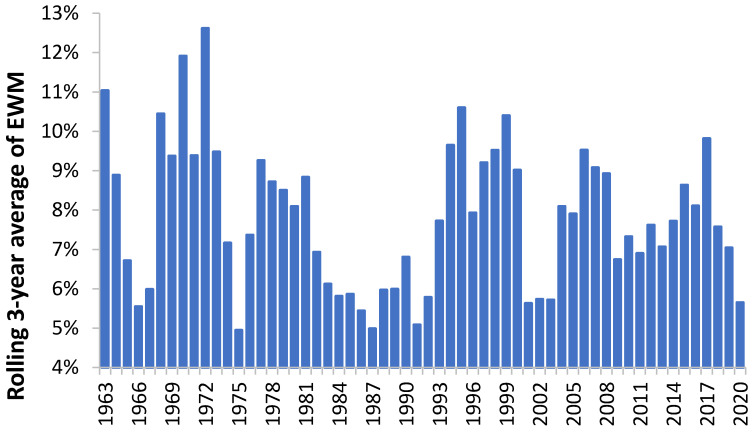
Rolling 3-year average of EWM in Singapore (1960 to 2020).

**Figure 8 ijerph-18-02161-f008:**
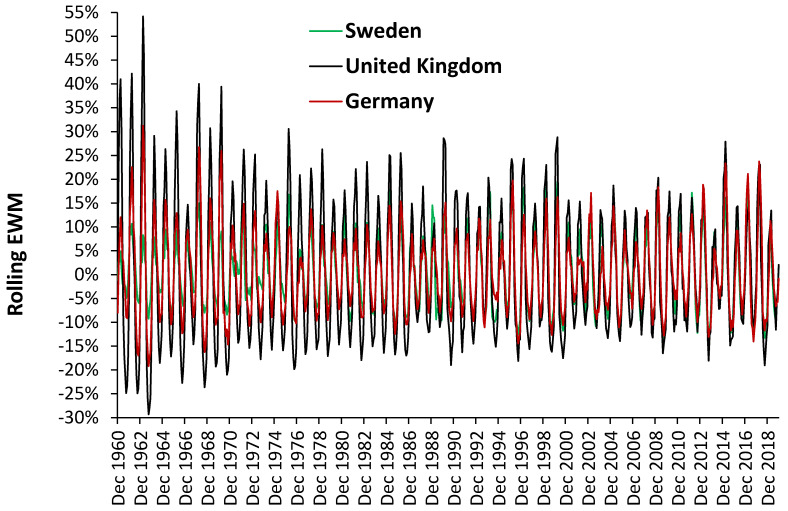
Rolling EWM in Sweden, Germany, and the United Kingdom (1960 to 2019).

**Figure 9 ijerph-18-02161-f009:**
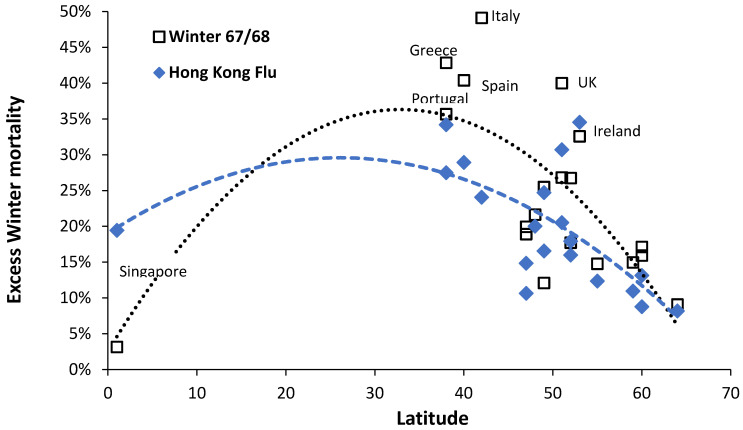
Excess winter mortality before and during the Hong Kong influenza pandemic. Black dotted line is line of best fit for winter of 1967/68, while blue dashed line is line of best fit for the pandemic year.

**Figure 10 ijerph-18-02161-f010:**
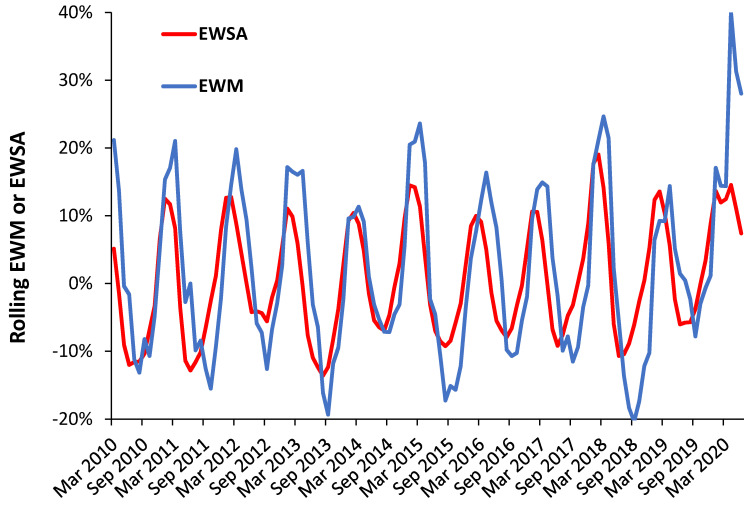
Rolling EWM and EWSA calculation in the Midlands of England. EWSA uses NHS staff sickness absence.

**Figure 11 ijerph-18-02161-f011:**
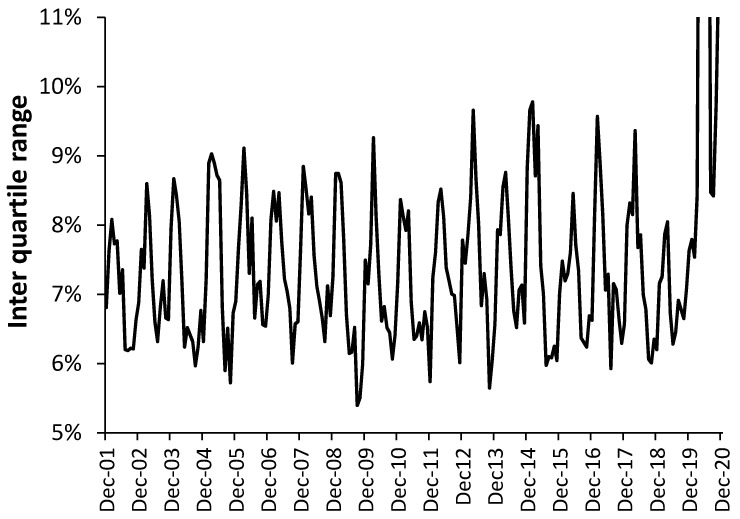
Inter quartile range arising from a rolling EWM calculation across 510 sub-national geographies (local authorities, counties, etc.) across the whole UK.

**Table 1 ijerph-18-02161-t001:** Maximum value of EWM (Excess Winter Mortality) in Michigan for various conditions (2000–2020), when the maximum occurred, and the proportion of total deaths due to that condition.

Variable	Pneum-Onia/Flu	Septic-Emia	COPD	Heart Disease	Stroke	All Other	Cancer	All Deaths
Maximum	103%	41%	40%	19%	19%	16%	8%	15%
When	April-18	March-13	April-05	March-05	March-20	March-05	January-04	March-05
Proportion	2%	1%	6%	27%	5%	36%	23%	100%

COPD = chronic obstructive pulmonary disease.

**Table 2 ijerph-18-02161-t002:** Maximum value of EWM in England and Wales (2010–2020) by age band using weekly data, plus the standard deviation associated with the rolling EWM calculation.

Variable	<1	01–14	15–44	45–64	65–74	75–84	>85
Maximum (M + F)	11.3%	24.4%	10.9%	15.5%	18.7%	26.7%	42.8%
Standard Deviation	5.5%	9.8%	4.2%	6.5%	8.0%	11.6%	16.8%
Male (maximum)	15.7%	25.6%	10.9%	14.0%	18.0%	24.5%	38.3%
Female (Maximum)	20.7%	42.5%	15.0%	17.7%	21.8%	30.0%	45.5%

**Table 3 ijerph-18-02161-t003:** Maximum value of EWM for regions in England and Wales (2010–2020) using weekly data, plus the standard deviation (STDEV) associated with the rolling EWM calculation, and the average annual deaths in each region.

Variable	North West	South East	Yorkshire & Humber	London	Wales	West Midlands	East	North East	East Midlands	South West
Annual deaths	69,548	78,470	50,580	48,387	32,249	52,659	54,535	27,032	43,140	54,681
Maximum EWM	27.6%	27.9%	28.2%	28.4%	28.8%	29.1%	30.1%	30.3%	30.4%	30.8%
STDEV	11.8%	12.3%	12.2%	11.5%	12.2%	12.2%	12.2%	11.8%	12.4%	11.6%
When	6 April 18	13 March 15	20 March 15	27 March 15	6 April 18	17 April 15	27 March 15	23 March 18	24 April 15	27 March 15

**Table 4 ijerph-18-02161-t004:** Maximum value of EWM for small areas ranked by deprivation decile in England and Wales (2005–2018) using weekly data, plus the time at which the rolling EWM (weekly data) reaches its maximum value.

Deprivation Decile	England		Wales	
	Maximum	When	Maximum	When
1 (most deprived)	26%	28-March-15	35%	28-April-18
2	28%	25-April-15	32%	04-April-15
3	29%	25-April-16	39%	07-April-18
4	29%	24-March-18	31%	14-April-18
5	31%	28-March-15	34%	28-April-18
6	30%	14-March-15	30%	24-March-18
7	29%	21-March-15	31%	02-May-15
8	29%	24-March-18	37%	25-April-15
9	30%	28-March-15	32%	07-April-18
10 (least deprived)	29%	24-March-18	28%	27-April-13
Total	29%	28-March-15	29%	28-April-18

**Table 5 ijerph-18-02161-t005:** European countries ranked according to the standard deviation associated with their rolling EWM (1960–2019).

Country	Standard Deviation of Rolling EWM
Finland, Norway, Iceland, Denmark, Sweden.	6.6% to 7.6%
Netherlands, Germany, Luxembourg, Switzerland, Austria, France, Belgium.	8.2% to 10.0%
Bulgaria, Italy, Greece.	12.1% to 13.4%
UK, Ireland.	13.9%
Spain.	14.5%
Portugal.	17.5%

**Table 6 ijerph-18-02161-t006:** Effect of place of death on EWM for the winter of 2017/18 in England and Wales, weekly data.

Parameter	Care Home	Communal Establishment	Hospital	Elsewhere	Home	Hospice
EWM	40.2%	31.2%	30.0%	20.5%	18.7%	7.8%
When	23 March 2018	13 April 2018	23 March 2018	4 May 2018	23 March 2018	9 February 2018
Proportion	22%	0.4%	47%	2%	24%	6%

## Data Availability

All data is publicly available. A copy of analysis can be obtained from the author on request.

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
