# Peer review of "Excess Winter Mortality (EWM) as a Dynamic Forensic Tool: Where, When, Which Conditions, Gender, Ethnicity and Age"

_ijerph, 2021, doi:10.3390/ijerph18042161_

Round 1
Reviewer 1 Report
Thank you for inviting me to review this interesting article. This manuscript investigate the dynamic issues behind international variation in Excess winter mortality (EWM). Generally, the paper is well written and the methods and results are interesting. This manuscript will be suitable for publication after revision.
- Abstract: The abstract needs more of a ‘hook’ to engage the reader and establish the novelty of the analysis in the scientific literature. An additional final sentence about the significance of these results or repercussions they would have on the community would make the abstract stronger.
- The Introduction section should be reorganized, provide additional information related to the methods followed in previous Studies. There is no relevant background into the methods used (at least in the introduction anyway), or statement of why this research is particularly novel aside from the study area. Maybe describe the relevance of the study and why the EWM analysis in this area is particularly unique.
- Clearly state the objective(s) in the last paragraph of the introduction.
- Are you able to access any level of socio-economic factors? Would it be possible to identify more economically-advantaged areas/countries from those that are not as advantaged? Given the large dataset you have, I think it would be possible to tease apart differences in SES and mortality (given the differences you describe in the discussion).
- Please, explain in the discussion the limitations of the work.
Figure captions:
- Figures 5 is blurry.
- Figure 6 is blurry.
- Figure 8 is blurry.
- Figure 9 is blurry.
Author Response
Many thanks for your comments and your time.
The abstract has been completely rewritten and major changes made to the Introduction.
The objectives are stated at the end of the Introduction.
Socio economic factors may need to wait for a future paper.
Limitations section has been added to the Discussion.
Alas the IJERPH template does strange things to figures when you attempt to resize them. Partly my mistake as the IJERPH had to change the margins which I had wrongly changed in the template.
After submitted the first draft I stumbled across some extra data from Michigan in the US and re-analysed my own data for the UK, hence two extra sections have been added, all highlighted in yellow.
Reviewer 2 Report
The author presents compelling examples to illustrate the utility of a rolling excess winter mortality calculation as an alternative to the traditional static method. The approach has merit and the examples show how the rolling EWM can be used to explore the influence of the social determinants of health and potential to inform policy and decision-making.
Unfortunately, I found the manuscript difficult to read. It would benefit from greater focus and organization. Specific details listed below. For example, in the Introduction (lines 59-61) the author writes, "This study will employ a rolling EWM calculation to assess EWM wider afield than just temperate areas, and especially investigate gender and detailed age differences in EWM." However, the datasets used within the results section vary. Some focus on England and Wales, others on Singapore, and others are global including data from 130 countries. So additional clarity in the can be added to describe that the paper presents selected examples drawn from the global dataset to illustrate particular topics.
Introduction - is jumbled and hard to follow.
Lines 34-61: What are the key aspects we should know from the four studies summarized? Can this section be integrated rather than making each study a separate paragraph?
Lines 35-37: Need a reference for the US study when it is introduced.
Materials and Methods
Line 66: Author should tell us more about the missing data. How much was imputed?
Is the author able to provide his spreadsheet model as supplemental data?
Results
Lines 123-126: Can be moved to Materials and Methods.
Figures 5, 6, 8, 10 have a Japanese character in the x-axis labels
Figure 6 has formatting and legend problems. y-axis label overlaps the numbers; lacks a legend for the red and purple lines
Line 233: Section 3.8 does not address the issue of the median as a stable measure. It does discuss variability and factors that contribute to EWM.
Line 289: Figure 9 has incorrect title
Line 291: This should be figure 10
Lines 304-306: Is there a reference for this statement
Line 331: "skin diseases"
Line 348: "over time"
Line 414: "residence"
Author Response
Many thanks for your time and valuable comments.
Both abstract and introduction have be rewritten. Hopefully the flow in the Introduction has been improved.
Reference to US study added at the start rather than the end.
Imputed data - simple answer very little and mostly smaller countries. No effect on the median EWM but allows the discontinuity each side of the missing year to be partly addressed.
Alas the IJERPH template seems to do strange things to the Figures when they get resized. Let me know if there are still issues. I may need to drop the Figures into PowerPoint and then save as JPEG before adding to the IJERPH template so resizing is fully scalable.
Skin and over time have been modified. You may need to clarify regarding halls of residence.
Title of section 3.8 changed to reflect the issues
Title to Figure 9 changed.
Extra references added.
After submitted the first draft I stumbled across some extra data from Michigan in the US and re-analysed my own data for the UK, hence two extra sections have been added, all highlighted in yellow. These expand upon some of the concepts.
Reviewer 3 Report
General
Applying a rolling calculation which deviates from the standard approach of the four winter months with March as a reference point against the other non-winter months the authors demonstrate that the maximum excess winter mortality (EWM) may occur in other months. Considering this allows comparison of data of countries situated in different latitudes as well as those of the southern globe. This rolling EWM assessment, which includes data from these different areas gives support to previous findings that cold is the predominant factor and that females are more susceptible than males.
Based on these data the impact of latitude, respiratory conditions, age and gender, location and area deprivation, ethnicity, influence on EWM is discussed. Quite interesting is the conclusion that while influenza outbreaks are obvious candidates for EWM a multitude of factors need to be considered, so hat vaccination in isolation does not seem to make a major impact due to the other winter pathogens.
Although the manuscript is convincing and seems to introduce the new aspect of rolling EWM, its novelty should be evaluated by an expert in this area of public health.
As such the manuscript is well designed, provides a great amount of information, from which appropriate conclusions have been drawn.
Smaller points
Line 226: Figure n?, possibly Fig. 6?
Table 4: Is the standard deviation of 8.2% to 10% attributed to Austria, France, Belgium? If so, insert a full stop after Switzerland.
Author Response
Many thanks for your comments and for spotting the smaller points. It is always useful to have a non-expert read the paper to see that it makes sense. Hopefully the Public Health community will spot the wider applicability of such a simple method. The issues in Line 266 and Table 4 have been addressed.
After submitted the first draft I stumbled across some extra data from Michigan in the US and re-analysed my own data for the UK, hence two extra sections have been added, all highlighted in yellow. These illustrate some of the concepts in greater detail.
Round 2
Reviewer 1 Report
I recommend the publication of the manuscript with these changes. I think it is very helpful to have extended the analysis to so many more locations than has been presented in other papers.